# SCENE-R1: VIDEO-GROUNDED LARGE LANGUAGE MODELS FOR 3D SCENE REASONING WITHOUT 3D ANNOTATIONS

## ABSTRACT

Currently, utilizing large language models to understand the 3D world is becoming popular. Yet existing 3D-aware LLMs act as black boxes: they output bounding boxes or textual answers without revealing how those decisions are made, and they still rely on pre-trained 3D detectors to supply object proposals. We introduce Scene-R1, a video-grounded framework that learns to reason about 3D scenes without any point-wise 3D instance supervision by pairing reinforcement-learning-driven reasoning with a two-stage grounding pipeline. In the temporal grounding stage, we explicitly reason about the video and select the video snippets most relevant to an open-ended query. In the subsequent image grounding stage, we analyze the image and predict the 2D bounding box. This 2D prediction is then refined into a precise 3D localization by matching against high-fidelity candidates from a zero-shot segmentation module, which captures fine geometry while eliminating the need for detector-based proposals. Scene-R1 can also adapt to the 3D visual question answering task to answer free-form questions directly from video. Our training pipeline only needs task-level 2D boxes or textual labels without dense 3D point-wise labels. Scene-R1 surpasses existing open-vocabulary baselines on multiple datasets, while delivering transparent, step-by-step rationales. These results show that reinforcement-learning-based reasoning combined with RGB-D video alone offers a practical, annotation-efficient route to trustworthy 3D scene understanding.

## 1 INTRODUCTION

Large language models (LLMs) have rapidly expanded beyond text, absorbing 2D visual perception and showing early promise in embodied AI and robotics applications Vemprala et al. (2023); Brohan et al. (2023). Extending these capabilities to real-world 3D scene understanding is a natural next step, and several recent works already tackle 3D visual grounding Chen et al. (2020); Wang et al. (2023); Achlioptas et al. (2020), captioning Chen et al. (2023; 2024; 2021), or question answering Azuma et al. (2022); Ma et al. (2023) directly on point clouds. Despite this progress, today's 3D-aware LLMs (3DLLMs) inherit two critical limitations, as shown in Figure 1 (a). First, their predictions are largely opaque: they output oriented boxes or short textual answers without exposing the intermediate chain of reasoning, making debugging and safety certification difficult. Second, they still depend on pre-trained 3D detectors or transformer-based instance segmenters that are themselves trained on dense point-wise labels, such as those using Mask3D Schult et al. (2023) trained with instance segmentation on ScanNet Dai et al. (2017). Acquiring such annotations remains costly and often infeasible for large-scale, in-the-wild RGB-D video.

Parallel advances in reinforcement-learning-driven reasoning offer a potential remedy. Group-Relative Policy Optimization (GRPO) and its open-source instantiation DeepSeek-R1 Guo et al. (2025) optimise LLMs to think aloud, producing detailed chains of thought and higher task accuracy without human-written rationales. Vision-R1 Huang et al. (2025) extends these ideas to images, confirming that purely RL-based objectives can endow multimodal models with transparent decision making. Yet, to date, no work leverages R1-style RL for video-level 3D perception or attempts to remove 3D instance supervision entirely.

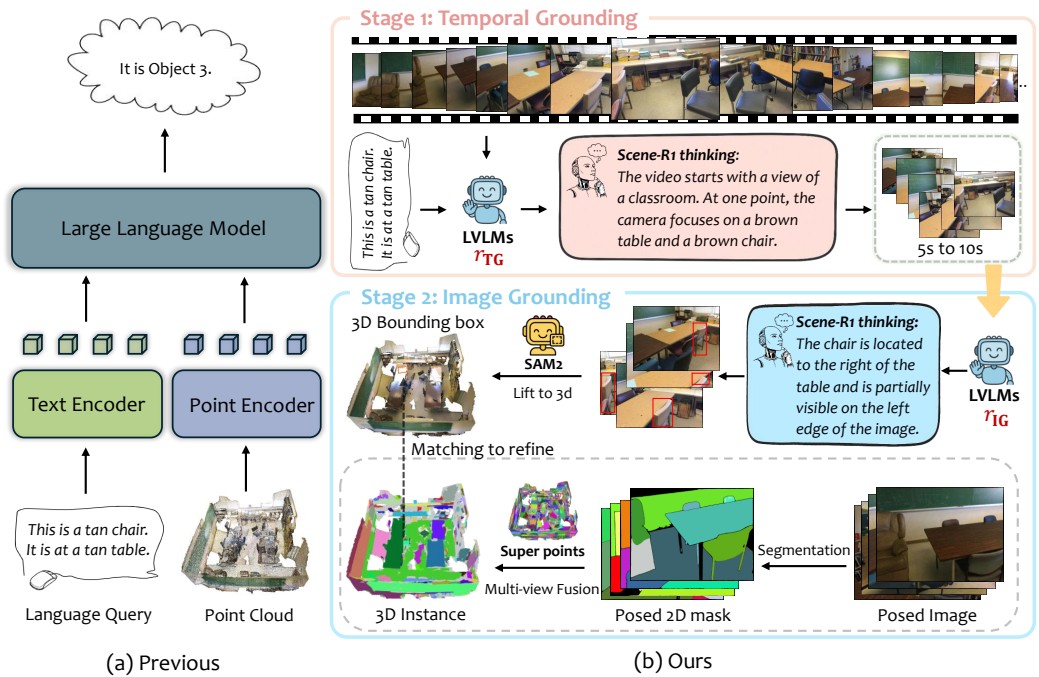

(a) Previous                    (b) Ours

Figure 1: **Overview of Scene-R1 *versus* prior 3D LLMs. (a)** Previous 3D LLMs directly output predictions as black boxes without exposing intermediate reasoning, while still depending on pretrained 3D encoders. **(b)** In contrast, Scene-R1 performs transparent 3D scene reasoning while bypassing the need for dense, point-wise 3D annotations.

We address this gap with Scene-R1, a video-grounded VLM that dispenses with explicit 3D labels and makes its reasoning process fully observable. As shown in Figure 1 (b), Scene-R1 is organized into two grounding stages that share the same VLM backbone but are fine-tuned under distinct RL objectives. In the temporal grounding stage, the vision language model selects query-relevant RGB snippets and generates explicit chain-of-thought rationales. In the subsequent image grounding stage, the VLM first grounds the target object by generating 2D bounding boxes in text format. To achieve a precise 3D localization, we introduce a two-step refinement process. The initial 2D prediction is first lifted into a 3D bounding box. Then it is refined by selecting the best-matching instance from a set of high-fidelity 3D candidates that are pre-generated by a module inspired by SAI3D Yin et al. (2024). This process of coarse localization followed by geometric refinement allows our model to capture fine geometry while bypassing dense point-wise supervision. Furthermore, our method can easily adapt to other tasks like 3D visual question answering by using the similarity between the predicted answers and ground truth labels as reward signals. Together, these advances show that video-centric perception, coupled with reinforcement-learning-based reasoning, offers a practical route to transparent and scalable 3D scene understanding without the cost of dense 3D annotation.

Our approach yields three central contributions:

- **RL-generated transparent reasoning.** Scene-R1 is, to our knowledge, the first 3D framework that generates and exposes chain-of-thought rationales through R1-style reinforcement learning, closing the interpretability gap left by prior 3DLLMs.

- **A novel two-stage, video-grounded pipeline.** Scene-R1 decomposes the complex 3D grounding task by first performing temporal grounding to reason about and select relevant video snippets, followed by an image grounding stage. This architecture uniquely enables end-to-end reasoning directly on video streams, bypassing the need for offline 3D scene reconstruction.

- **Annotation-efficient 3D grounding.** By combining our RL-driven, two-stage pipeline, Scene-R1 learns effectively from lightweight 2D rewards, yet achieves performance competitive with detector-based pipelines on standard 3D indoor understanding benchmarks.

## 2 RELATED WORK

### 2.1 3D INDOOR SCENE UNDERSTANDING

3D indoor scene understanding enables embodied agents to perceive and interpret spatial structures within indoor environments. This capability supports a range of downstream tasks, including: 1) 3D visual Grounding Chen et al. (2020); Achlioptas et al. (2020); Wang et al. (2023) for language-based object localization; 2) 3D Dense Captioning Chen et al. (2023; 2021) for generating object-level descriptions; 3) 3D Visual Question Answering Azuma et al. (2022); Ma et al. (2023) for answering questions grounded in 3D context; and 4) Affordance grounding Delitzas et al. (2024) for predicting human-object interactions. While early works Ma et al. (2023); Azuma et al. (2022); Chen et al. (2020); Achlioptas et al. (2020) often focus on a single task, recent research has developed unified models Huang et al. (2023; 2024) for more versatile usage.

### 2.2 MULTI-MODAL LARGE LANGUAGE MODELS

Multi-modal large language models (MLLMs) enhance LLMs by integrating inputs from modalities like images Radford et al. (2021); Rombach et al. (2021), video Li et al. (2023); Wang et al. (2024), and 3D point clouds Peng et al. (2023); Fu et al. (2024); Guo et al. (2023). In the 3D realm, LERF Kerr et al. (2023) learns a language field inside NeRF but requires per-scene optimization. Openscene Peng et al. (2023) unifies segmentation and grounding by distilling from 2D vision-language models. Chat-scene Huang et al. (2024) uses object-centric representations for object-level reasoning within the language model. In contrast to these approaches, our method operates directly on source videos, eliminating the need for 3D asset inputs and costly 3D annotations.

### 2.3 REINFORCEMENT LEARNING

With increasing attention to model trustworthiness Guo et al. (2025); OpenAI (2024), recent works leverage reinforcement learning (RL) to enhance the reasoning capabilities of VLMs. R1-V Chen et al. (2025) shows that LVLMs trained with RL exhibit improved generalization on image reasoning tasks. Timezero Wang et al. (2025) employs RL for accurate temporal frame grounding, while Video-R1 Feng et al. (2025) encourages models to leverage temporal information by contrasting performance on ordered versus shuffled frames. However, leveraging R1-style RL to generate transparent, chain-of-thought reasoning for 3D indoor scene understanding remains an underexplored area, which this work addresses.

## 3 PRELIMINARY

### 3.1 3D LARGE LANGUAGE MODELS

Contemporary 3D-aware language–vision models follow a shared, instance-centric pipeline, as shown in Figure 1 (a). A point-cloud object detector—typically an instance segmentation network such as Mask3D Schult et al. (2023)—constitutes the critical first step: it decomposes the raw scene into $n$ object masks $\{\mathcal{P}_i\}_{i=1}^n$. For every object, a 3D encoder (e.g., Uni3D Zhou et al. (2023)) extracts geometric features, while a 2D encoder captures appearance cues from multi-view projections. Lightweight visual language projectors map 3D and 2D features to the language token space. Concatenating the identifier, 3D and 2D embeddings yields a scene-level token sequence in which the LLM attends to ground queries and produces answers. Because the opening object-detector stage demands dense point-wise instance masks for supervision, the entire pipeline inherits high annotation costs and remains tied to a handful of curated datasets. Scene-R1 removes this dependency by discarding the detector and learning directly from RGB-D video without any 3D instance labels.

### 3.2 DEEPSEEK-R1 AND GROUP-RELATIVE POLICY OPTIMIZATION

DeepSeek-R1 Guo et al. (2025) shows that an LLM can be post-trained *entirely* by reinforcement learning, eliminating the supervised fine-tuning stage. Its optimization backbone, *Group-Relative Policy Optimization* (**GRPO**) Shao et al. (2024) departs from actor–critic methods such as PPO Schulman et al. (2017) in two respects.

**Sampling and group-normalized reward.** For every prompt $p$ the current policy $\pi_\theta$ generates a *group* of $G$ candidate responses $\mathbf{o} = \{o_1, \ldots, o_G\}$. A task-specific scalar reward function $r(\cdot)$ is applied to each response, yielding $\{r(o_i)\}_{i=1}^G$. GRPO converts raw rewards into *relative* scores.

$$\bar{r}(o_i) = \frac{r(o_i) - \mu}{\sigma}, \quad \mu = \tfrac{1}{G} \sum_{j=1}^G r(o_j), \; \sigma = \sqrt{\tfrac{1}{G} \sum_{j=1}^G (r(o_j) - \mu)^2}. \tag{1}$$

The group-wise standardization makes the update invariant to the reward scale and emphasizes responses that outperform their peers.

**Weighted objective.** Let $\pi_{\theta_{\text{old}}}$ denote the policy parameters from the previous optimization step. The group-normalised rewards are combined with an importance-sampling ratio to form

$$R(\mathbf{o}) = \sum_{i=1}^G \frac{\pi_\theta(o_i)}{\pi_{\theta_{\text{old}}}(o_i)} \, \bar{r}(o_i), \tag{2}$$

where $\pi_\theta(o_i)$ is the probability assigned by the current policy to response $o_i$. Training maximises the KL-regularised objective

$$\max_\theta \; \mathbb{E}_{\mathbf{o} \sim \pi_{\theta_{\text{old}}}(p)} \Big[ R(\mathbf{o}) - \beta \, D_{\text{KL}}\big(\pi_\theta \,\|\, \pi_{\text{ref}}\big) \Big], \tag{3}$$

where $\pi_{\text{ref}}$ is the frozen reference model and $\beta$ controls the trust-region size. Crucially, Eq. equation 3 obviates the need for a learned critic, reducing memory overhead and stabilising optimization when reward computation—e.g. IoU over long videos—is costly. Scale-invariance of equation 2 permits mixing heterogeneous rewards (frame-IoU, box-IoU, exact-match) under a single learning rate schedule.

## 4 METHOD

We address the task of grounding free-form textual queries in RGB-D video and reasoning about the referred 3D regions without using point-wise 3D annotations, as shown in Figure 1 (b). Formally, each scan have a coresponding video is $\mathcal{V} = \{(I_t, D_t, \mathbf{T}_t)\}_{t=1}^T$, where $I_t$ and $D_t$ are color and depth images and $\mathbf{T}_t$ the camera pose; $\mathbf{K}$ denotes fixed intrinsics. Given a query $q$, the model must (i) return the 3D region that $q$ describes, (ii) answer any accompanying question, and (iii) supply a chain-of-thought explanation. Scene-R1 is built on the publicly released Qwen2.5-VL Bai et al. (2025) backbone, whose instruction tuning already encompasses 2D grounding and general VQA. We exploit this prior to teach it to understand the 3D world and minimise the amount of task-specific reinforcement learning (RL). The same architecture is used in all tasks optimized with GRPO.

### 4.1 STAGE 1: TEMPORAL GROUNDING

For 3D visual grounding and affordance grounding, we employ a two-stage pipeline. Given a natural-language description, the model first predicts a continuous time window $\hat{S} = [\hat{t}_s, \; \hat{t}_e]$ which should enclose all frames containing the target object. Let fps be the video frame rate. The window endpoints are converted to integer frame indices,

$$\hat{f}_s = \lfloor \hat{t}_s \cdot \text{fps} \rfloor, \qquad \hat{f}_e = \lceil \hat{t}_e \cdot \text{fps} \rceil,$$

yielding the predicted frame set $\hat{\mathcal{L}} = \{\hat{f}_s, \; \hat{f}_s + 1, \ldots, \hat{f}_e\}$. The ground-truth set of relevant frames is $\mathcal{L}^\star = \{\ell_1, \ldots, \ell_{|\mathcal{L}^\star|}\} \subset \{1, \ldots, T\}$. We measure overlap via the frame-level Intersection-over-Union

$$\text{IoU}_{\text{time}}(\hat{\mathcal{L}}, \mathcal{L}^\star) = \frac{|\hat{\mathcal{L}} \cap \mathcal{L}^\star|}{|\hat{\mathcal{L}} \cup \mathcal{L}^\star|}.$$

In addition, answers must follow the template `<think>...</think> <answer> <`$t_s$`> to <`$t_e$`></answer>` to first output the chain of thoughts, then the answers. We therefore define a format reward

$$r_{\text{form}}(o) = \begin{cases} 1, & \text{if the output } o \text{ matches the template;} \\ 0, & \text{otherwise.} \end{cases}$$

and pass to GRPO the combined reward

$$r_{\text{TG}} = \text{IoU}_{\text{time}}(\hat{\mathcal{L}}, \mathcal{L}^{\star}) \; + \; \lambda \, r_{\text{form}}(o), \qquad \lambda = 0.1.$$

The resulting window $\hat{S}$ along with the corresponding RGB-D frames is then forwarded to Stage 2 for image-level object localization.

## 4.2 STAGE 2: IMAGE GROUNDING

Conditioned on the temporal window $\hat{S}$ obtained in Stage 1, the model now localizes the target object in every retained frame. For each frame index $\tau \in \hat{S}$, the Qwen2.5-VL decoder is prompted to output a JSON line: {"bbox_2d": [x₁, y₁, x₂, y₂], "label": <object-class>}, where $(x_1, y_1)$ and $(x_2, y_2)$ denote the top-left and bottom-right pixel coordinates of the predicted box $\hat{\mathbf{b}}_\tau = (x_1, y_1, x_2, y_2)$. Given the ground-truth box $\mathbf{b}_\tau^{\star}$ for frame $\tau$, we compute the spatial Intersection-over-Union

$$\text{IoU}_{\text{box}}(\hat{\mathbf{b}}_\tau, \mathbf{b}_\tau^{\star}) = \frac{|\hat{\mathbf{b}}_\tau \cap \mathbf{b}_\tau^{\star}|}{|\hat{\mathbf{b}}_\tau \cup \mathbf{b}_\tau^{\star}|}.$$

We reuse the think/answer format reward from Stage 1, and introduce an additional JSON-specific reward

$$r_{\text{json}}(o) = \begin{cases} 1, & \text{valid JSON with the schema above;} \\ 0, & \text{otherwise.} \end{cases}$$

The final reward pass to GRPO is

$$r_{\text{IG}} = \text{IoU}_{\text{box}}(\hat{\mathbf{b}}_\tau, \mathbf{b}_\tau^{\star}) \; + \; \lambda \big( r_{\text{json}}(o) \; + \; r_{\text{form}}(o) \big).$$

This design incentivises the model to produce accurate spatial localizations, its chain of thought, and adhere strictly to the required JSON schema.

## 4.3 LIFTING 2D PREDICTIONS TO 3D

To obtain a true 3D localization, we turn each frame-wise box into a dense mask, back-project the masked pixels, and unite the resulting points across the entire temporal window $\hat{S}$. For every predicted box $\hat{\mathbf{b}}_\tau$ at frame $\tau$, we feed the RGB image and the box coordinates to SAM2 Ravi et al. (2024) as a prompt. SAM2 propagates the cue over neighbouring frames, yielding a binary mask $M_\tau \in \{0, 1\}^{H \times W}$ where $M_\tau(u, v) = 1$ marks object pixels. Each foreground pixel $(u, v)$ with depth $D_\tau(u, v)$ is converted to a 3D point in the camera coordinate system,

$$\mathbf{x}_{\text{cam}} = D_\tau(u, v) \, \mathbf{K}^{-1} \begin{bmatrix} u \\ v \\ 1 \end{bmatrix},$$

where $\mathbf{K}$ is the intrinsic matrix. The point is then lifted into the world coordinates via the camera pose $\mathbf{T}_\tau \in \mathbb{R}^{4 \times 4}$:

$$\mathbf{X} = \mathbf{T}_\tau \begin{bmatrix} \mathbf{x}_{\text{cam}} \\ 1 \end{bmatrix}.$$

Aggregating all such points over $\tau \in \hat{S}$ produces the instance-level point cloud

$$\mathbf{P} \; = \; \bigcup_{\tau \in \hat{S}} \{ \mathbf{X} \mid M_\tau(u, v) = 1 \}.$$

While the tightest axis-aligned bounding box around $\mathbf{P}$ provides an initial 3D localization, it often lacks geometric fidelity. To generate a more precise output, we introduce a final refinement step inspired by the zero-shot 3D instance segmentation method, SAI3D Yin et al. (2024). Specifically, we leverage its approach of merging geometric primitives based on multi-view 2D mask consistency to transform our initial point cloud $\mathbf{P}$ into a high-quality voxelized instance mask. This allows us to significantly improve the geometric accuracy of our final 3D output while remaining fully consistent with our framework's core principle: operating entirely without 3D instance supervision.

### 4.4 3D Visual Question Answering

Scene-R1 can be fine-tuned for 3-D visual question answering (3DVQA) with only minor changes to the reinforcement objective. The `<think>` block provides the chain of thought, while the `<answer>` tag encloses the final answer $a$. Let $a^\star$ denote the ground-truth answer. We adopt two complementary metrics from Huang et al. (2023):

- Exact Match (EM): $\text{EM}(a, a^\star) = 1$ iff the normalised strings are identical.
- Refined Exact Match (EM-R): a soft variant that tolerates minor lexical variation.

We reuse the `think`/`answer` formatting reward $r_{\text{form}}(o)$ introduced in Stage 1. The scalar reward supplied to GRPO is

$$r_{\text{QA}} = \text{EM}(a, a^\star) + \text{EM-R}(a, a^\star) + \lambda\, r_{\text{form}}(o).$$

This one-stage formulation rewards the model directly for factual accuracy and explanatory clarity, enabling Scene-R1 to answer 3D questions without ever consulting dense 3D annotations.

## 5 EXPERIMENTS

This section assesses Scene-R1 on three core 3D scene understanding tasks: visual grounding, affordance grounding, and visual question answering. We first describe implementation details and evaluation protocols (Sec. 5.1), then present quantitative (Sec. 5.2) and qualitative (Sec. 5.3) results, followed by an ablation study that analyses design choices (Sec. 5.4).

### 5.1 IMPLEMENTATION DETAILS

**Datasets.** ScanRefer Chen et al. (2020) extends ScanNet Dai et al. (2017) RGB-D reconstructions with free-form referring expressions that localize objects directly in 3D space. The corpus contains 51,583 descriptions referring to 11,046 target objects across 800 indoor scenes Dai et al. (2017). SceneFun3D Delitzas et al. (2024) targets fine-grained functionality understanding in real-world indoor scans captured with a high-resolution Faro laser scanner Baruch et al. (2021). VSI-Bench Yang et al. (2025) is a video-based benchmark designed to evaluate the visual-spatial intelligence of Multimodal Large Language Models (MLLMs). It includes over 5,000 question-answer pairs derived from 288 real-world indoor videos sourced from ScanNet, ScanNet++, and ARKitScenes. The benchmark features eight distinct tasks, such as object counting, route planning, and distance estimation, which test an MLLM's ability to see, remember, and recall spatial information from sequential visual input.

**Training schedule.** We initialise all experiments from the `Qwen2.5-VL-7B` checkpoint Bai et al. (2025). The model ingests a sequence of interleaved visual and textual tokens. RGB frames are sampled at 2 fps; ScanNet clips are resized to $640\times480$ while ARKitScenes clips retain their native resolution. To bound the computational footprint across heterogeneous inputs we employ the *smart-resize* heuristic of Bai et al. (2025): each frame is isotropically scaled so that the product "#frames $\times$ pixels" does not exceed $16384\times28\times28$ ($\approx$12.8 M pixels). All model weights are fine-tuned with the reinforcement-learning objectives described in Section 4. Optimization uses AdamW (learning-rate $1\times10^{-5}$, $\beta_1{=}0.9$, $\beta_2{=}0.95$, weight-decay 0.02) and a cosine decay schedule without warm-up. Training is distributed over 4 A100 80 GB GPUs in bfloat16 precision with fully-sharded data-parallelism; gradient accumulation yields an effective batch of 32 video–query pairs. A single pass over the training split (*one epoch*) suffices to converge both temporal- and spatial-grounding stages, requiring roughly 280 GPU-hours end-to-end.

### 5.2 QUANTITATIVE ANALYSIS

**3D Visual Grounding.** Table 1 summarizes ScanRefer Chen et al. (2020) grounding accuracy measured as the percentage of predictions whose 3D IoU with the ground-truth box exceeds 0.25 or 0.50. The upper block of the table lists fully-supervised methods that are trained with dense instance masks and currently define the performance ceiling; the middle block contains systems that forego grounding annotations but still depend on a detector or segmentor pretrained with point-wise

Table 1: 3DVG results on ScanRefer validation set. The accuracy on the "unique" subset, "multiple" subset, and whole validation set is all provided. Following Chen et al. (2020), we label the query as "unique" if it only contains a single object of its class. Otherwise, we label it as "multiple".

| Methods | Agent | Unique | | Multiple | | Overall | |
|---|---|---|---|---|---|---|---|
| | | Acc@0.25 | Acc@0.5 | Acc@0.25 | Acc@0.5 | Acc@0.25 | Acc@0.5 |
| **Fully Supervised Methods** | | | | | | | |
| ScanRefer Chen et al. (2020) | – | 65.0 | 43.3 | 30.6 | 19.8 | 37.3 | 24.3 |
| TGNN Huang et al. (2021) | – | 64.5 | 53.0 | 27.0 | 21.9 | 34.3 | 29.7 |
| InstanceRefer Yuan et al. (2021) | – | 77.5 | 66.8 | 31.3 | 24.8 | 40.2 | 32.9 |
| 3DVG-Transformer Zhao et al. (2021) | – | 81.9 | 60.6 | 39.3 | 28.4 | 47.6 | 34.7 |
| BUTD-DETR Jain et al. (2022) | – | 84.2 | 66.3 | 46.6 | 35.1 | 52.2 | 39.8 |
| ConcreteNet Unal et al. (2024) | – | 86.3 | 82.1 | 42.4 | 38.4 | 50.6 | 46.5 |
| **LLMs w/ 3D Inst. Supervision Dependency** | | | | | | | |
| ZSVG3D Yuan et al. (2024) | GPT-4 turbo | 63.8 | 58.4 | 27.7 | 24.6 | 36.4 | 32.7 |
| SeeGround Li et al. (2024) | Qwen2-VL-72b | 75.7 | 68.9 | 34.0 | 30.0 | 44.1 | 39.4 |
| LLaVA-3D Zhu et al. (2024) | LLaVA-Video-7B | – | – | – | – | 54.1 | 42.4 |
| Video-3D LLM Zheng et al. (2025) | LLaVA-Video-7B | – | – | – | – | 58.1 | 51.7 |
| 3D-LLaVA Deng et al. (2025) | LLaVA-1.5-7B | – | – | – | – | 51.2 | 40.6 |
| **Free from 3D Instance or Annotations Supervision** | | | | | | | |
| LERF Kerr et al. (2023) | CLIP | - | - | - | - | 4.8 | 0.9 |
| OpenScene Peng et al. (2023) | CLIP | 20.1 | 13.1 | 11.1 | 4.4 | 13.2 | 6.5 |
| VLM-Grounder Xu et al. (2024) | GPT-4o | **66.0** | 29.8 | 48.3 | 33.5 | 51.6 | 32.8 |
| **Ours (Scene-R1)** | Qwen2.5-VL-7B | 64.1 | **49.1** | **50.4** | **39.4** | **53.1** | **41.2** |

Table 3: Detailed performance comparison on VSI-Bench Yang et al. (2025). Models are grouped by their category.

| Methods | Numerical Answer | | | | Multiple-Choice Answer | | | | Avg. |
|---|---|---|---|---|---|---|---|---|---|
| | Obj. Count | Abs. Dist. | Obj. Size | Room Size | Rel. Dist. | Rel. Dir. | Route Plan | Appr. Order | |
| GPT-4o | 46.2 | 5.3 | 43.8 | 38.2 | 37.0 | 41.3 | 31.5 | 28.5 | 34.0 |
| Gemini-1.5 Pro | 56.2 | 30.9 | 64.1 | 43.6 | **51.3** | **46.3** | **36.0** | 34.6 | **45.4** |
| LongVA-7B | 38.0 | 16.6 | 38.9 | 22.2 | 33.1 | 43.3 | 25.4 | 15.7 | 29.2 |
| LLaVA-OneVision-7B | 47.7 | 20.2 | 47.4 | 12.3 | 42.5 | 35.2 | 29.4 | 24.4 | 32.4 |
| LLaVA-Video-7B | 48.5 | 14.0 | 47.8 | 24.2 | 43.5 | 42.4 | 34.0 | 30.6 | 35.6 |
| InternVL2-8B | 31.3 | 29.0 | 48.9 | 44.2 | 38.0 | 33.4 | 28.9 | 46.4 | 37.5 |
| LLaVA-OneVision-72B | 43.5 | 23.9 | 57.6 | 37.5 | 42.5 | 39.9 | 32.5 | 44.6 | 40.2 |
| LLaVA-Video-72B | 48.9 | 22.8 | 57.4 | 35.3 | 42.4 | 36.7 | 35.0 | **48.6** | 40.9 |
| Video-R1 Feng et al. (2025) | 47.0 | 46.1 | 64.2 | 46.5 | 29.1 | 26.7 | 30.9 | 11.7 | 37.8 |
| **Ours (Scene-R1)** | **60.8** | **46.7** | **66.0** | **46.9** | 29.0 | 35.9 | 28.9 | 17.0 | **41.4** |

3-D labels to supply proposal boxes and features; the lower block gathers approaches that operate without *any* 3-D supervision. In this strictest setting, Scene-R1 surpasses the previous best label-free baseline, VLM-Grounder Xu et al. (2024), by +1.5 and +8.4 percentage points, respectively. While fully-supervised detectors remain ahead in absolute terms, Scene-R1 demonstrates that a single video-grounded vision–language model, trained end-to-end with lightweight IoU rewards, can deliver competitive 3-D localization without specialised 3-D modules or point-cloud annotations, thereby offering a practical route toward scalable 3-D scene understanding.

**Task-driven Affordance Grounding.** Results on SceneFun3D Delitzas et al. (2024) are reported in Table 2 using mean average precision at IoU thresholds 0.50 and 0.25 ($AP_{50}$ and $AP_{25}$) over points. The fully supervised upper-bound, OpenMask3D-F, attains 8.0 / 17.5. When supervision is removed, the detector-free OpenMask3D baseline collapses to 0.0, while the LERF recovers 4.9 / 11.3. Scene-R1 improves this unsupervised state of the art to 12.0 $AP_{25}$,

Table 2: Quantitative results on task-driven affordance grounding.

| Methods | Supervision | $AP_{50}$ | $AP_{25}$ |
|---|---|---|---|
| OpenMask3D-F | fully | 8.0 | 17.5 |
| OpenMask3D | – | 0.0 | 0.0 |
| LERF | – | 4.9 | 11.3 |
| **Ours (Scene-R1)** | – | **6.3** | **12.0** |

corresponding to relative gains 6 % over LERF. Although a margin remains for the fully supervised detector, these numbers confirm that our video-grounded VLM can localize actionable regions from task descriptions with no geometric labels, further illustrating the versatility of the proposed annotation-free pipeline.

**3D Question Answering.** We evaluate our model's 3D question answering capabilities on the video-centric benchmark VSI-Bench. As shown in Table 3, we compare Scene-R1 against a series of

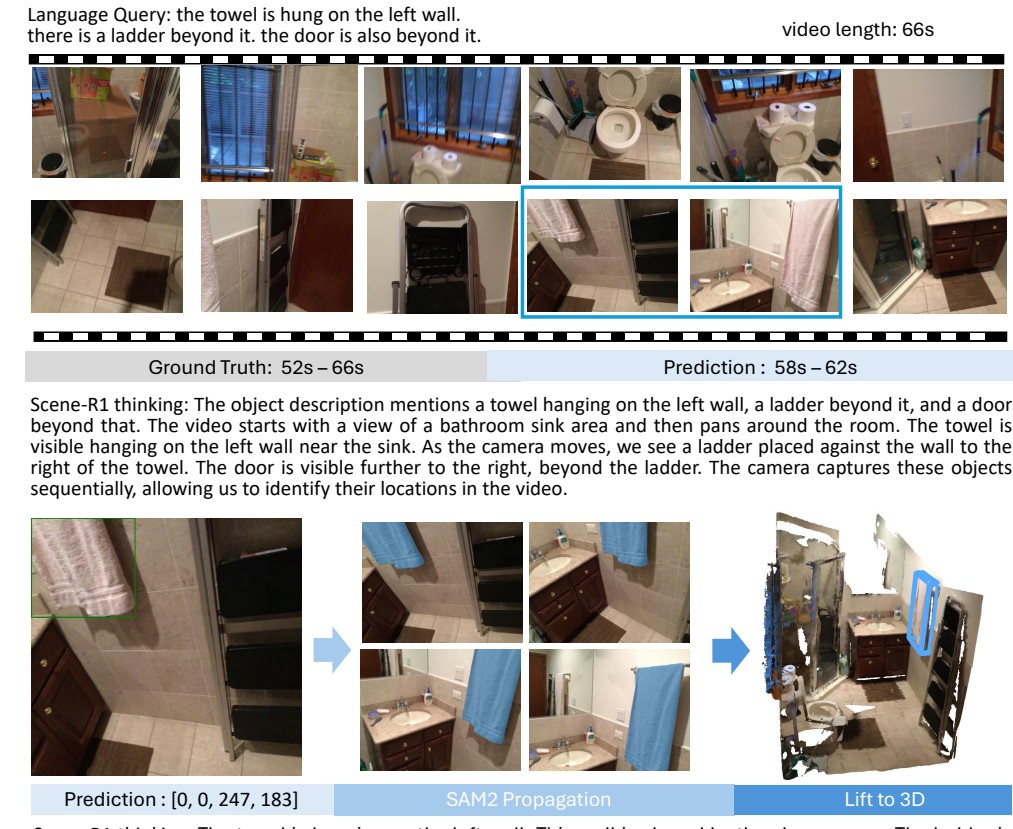

Language Query: the towel is hung on the left wall. there is a ladder beyond it. the door is also beyond it.

video length: 66s

Stage 1

Ground Truth: 52s – 66s          Prediction : 58s – 62s

Scene-R1 thinking: The object description mentions a towel hanging on the left wall, a ladder beyond it, and a door beyond that. The video starts with a view of a bathroom sink area and then pans around the room. The towel is visible hanging on the left wall near the sink. As the camera moves, we see a ladder placed against the wall to the right of the towel. The door is visible further to the right, beyond the ladder. The camera captures these objects sequentially, allowing us to identify their locations in the video.

Stage 2

Prediction : [0, 0, 247, 183]          SAM2 Propagation          Lift to 3D

Scene-R1 thinking: The towel is hanging on the left wall. This wall is shared by the shower area. The ladder is located on this wall, to the right of the towel. The door is also beyond the towel and the ladder, indicating it is farther back in the image.

Figure 2: Visualization of visual grounding.

models, where our method achieves a strong average performance of 41.4. Under the weak rewards described in Section 4, this excellent performance on a benchmark designed for its native video-stream modality further highlights Scene-R1's effectiveness. The results underscore our model's robust capabilities in handling complex 3D question answering and its emergent visual-spatial intelligence. This finding demonstrates that reinforcement learning with accuracy and format rewards is sufficient to equip a generic vision-language model with competitive 3D reasoning ability, without recourse to point-cloud annotations or task-specific architectural components.

## 5.3 QUALITATIVE ANALYSIS

**3D Visual grounding.** In Figure 2 we show a 3DVG example using our method. *Stage 1* (top) illustrates the temporal grounding stage. Scene-R1 selects frames 58–62s (blue bar) out of a 66s clip; this window covers all three referenced objects and overlaps the ground-truth segment (52s–66s) with a temporal IoU of 0.71. *Step 2* (bottom) shows the second stage: image grounding. The predicted 2D box (green) tightly encloses the towel, while the lifted 3D bounding box (blue mask) aligns well with the labelled towel region in the reconstructed point cloud. The generated chain-of-thought explains the spatial hierarchy (*"ladder beyond the towel, door further back"*) in natural language, providing human-readable evidence for the model's decision.

**Affordance grounding.** Figure 3 visualises two task-driven instructions. *Dial a number on the telephone on the nightstand*. Scene-R1 highlights the telephone keypad—precisely the actionable sub-region—within both the RGB frame and the 3D cloud, showing that it can attend to fine-grained keypoints rather than the entire object. *Open the bottom closet drawer between the door and the closet*. The model grounds the correct drawer despite multiple similar fronts and justifies the choice in its CoT

Table 4: Ablation study on stage 1: temporal grounding (left) and stage 2: image grounding (right).

| Model | Temporal Grounding | | | ScanRefer | | SceneFun3D | |
|---|---|---|---|---|---|---|---|
| | mIoU | R1@0.3 | R1@0.5 | Acc@0.25 | Acc@0.5 | Acc@0.25 | Acc@0.5 |
| QWen2.5-VL (zero-shot) | 15.1 | 13.4 | 3.8 | 32.2 | 25.9 | 20.3 | 2.5 |
| QWen2.5-VL (SFT) | 25.4 | 36.3 | 11.0 | 51.7 | 43.1 | 42.9 | 10.9 |
| VLM-Grounder Xu et al. (2024) | - | - | - | 62.4 | 53.2 | - | - |
| Ours w/o thinking | 33.1 | 54.8 | 23.4 | 68.6 | 57.3 | 46.5 | 9.1 |
| Ours | **34.3** | **56.6** | **24.8** | **70.4** | **59.4** | **48.5** | **11.0** |

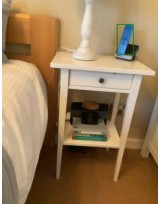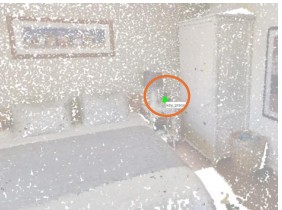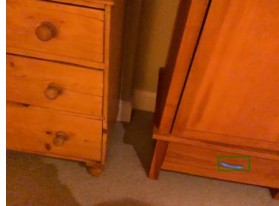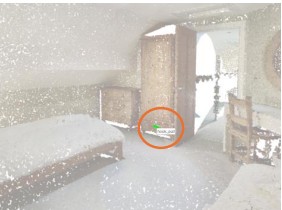

**Task:** Dial a number on the telephone on the nightstand

**Scene-R1:** In the image, there is a telephone on the nightstand. The telephone appears to be a cordless model with a keypad and a handset. The keypoint of interest for this task would be the keypad, as it is used to dial numbers.

**Task:** Open the bottom closet drawer between the door and the cabinet

**Scene-R1:** The instruction refers to opening a drawer that is located below a door, which is part of a larger piece of furniture that includes a cabinet. In the image, there is a wooden dresser with three drawers on the left side and a door on the right side. The bottom drawer is the one closest to the door and the cabinet. To open this drawer, you would need to pull it outwards from its current position.

Figure 3: Visualization of task-driven affordance grounding. For each instruction, we show the input RGB view (left) and the reconstructed 3-D scene (right) with the predicted affordance keypoint highlighted (green marker, circled in red). The accompanying natural-language rationales (below each pair) reveal how the model reasons about the constraints.

by referencing the relative position between the door and the cabinet. Across examples, Scene-R1 maintains spatial consistency between its text rationale, 2D output, and lifted 3D mask—illustrating how the unified pipeline of Sec. 4 yields interpretable, cross-modal alignment without any point-wise supervision.

### 5.4 ABLATION STUDY

As summarised in Table 4, reinforcement-learning-based optimization and explicit chain-of-thought prompting are both indispensable to Scene-R1's performance. On temporal grounding, GRPO lifts the backbone from 15.1 mIoU and 13.4 R1@0.3 in the zero-shot setting to 34.3 mIoU and 56.6 R1@0.3. Simply fine-tuning with cross-entropy (25.4 mIoU) or removing the `<think>` prompt (row "w/o thinking") yields far lower scores, confirming that RL and visible reasoning are both critical for tight temporal localization. The same pattern holds for image grounding: our full model attains 70.4 / 59.4 % Acc@0.25 / 0.5 on ScanRefer and 26.4 / 4.6 % on SceneFun3D—more than doubling the zero-shot backbone (32.2 / 25.9 % and 20.3 / 2.5 %, respectively) and dwarfing SFT (51.7 / 43.1 % and 42.9 / 10.9 %). Removing the thinking prompt again degrades results, underscoring that articulating the reasoning path helps the model attend to the correct pixels and 3-D regions. Together, these ablations show that GRPO and chain-of-thought supervision synergistically drive Scene-R1's gains in both temporal and spatial grounding.

### 6 CONCLUSION

We introduced Scene-R1, a video-grounded framework that performs explicit 3D scene reasoning without relying on point-wise 3D annotations. We decompose the task into a temporally-aware grounding stage and a spatially-refined segmentation stage, both optimized with critic-free, group-relative policy optimization and guided solely by lightweight *IoU* and format rewards. Scene-R1 sidesteps the proposal networks and dense point-cloud supervision that previous 3D LLM pipelines require. It produces faithful, interpretable chains of thought and shows that RGB-D video, paired with reinforcement-learning post-training, offers a practical and annotation-efficient route to holistic 3D scene understanding.

**Reproducibility Statement** We are committed to reproducibility. Our core source code is provided as a `.zip` file in the supplementary materials, which includes a `README.pdf` with setup instructions. An anonymous project page with a demo and visualizations is also available; its link can be found in a PDF file within the supplementary submission to ensure anonymity, in accordance with the review policy. The full, well-documented codebase will be publicly released upon publication.

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

# A    PROMPTS

To prompt the VLM to generate proporate output, we use different prompts for each task. For video grounding:

> To accurately pinpoint the object described as "[EVENT]" in the video, determine the precise time period of the occurrence of the object.
> Provide the start and end times (in seconds, precise to one decimal place) in the format "start time to end time" within the <answer> </answer> tags. For example: "12.5 to 17.0".

For image grounding:

> Outline the object according to the description "[EVENT]". Output the thinking process in <think> </think>. Outline the bbox_2d coordinates in JSON format.

For affordance grounding:

> Outline the functional interactive element referred to by the task description "[EVENT]". (e.g., a button affords pressing, a drawer knob affords pulling).

We replace the [EVENT] with the actual description in dataset.

# B    VISUALIZATIONS

Here we present two additional qualitative results that highlight Scene-R1's versatility across distinct 3-D reasoning tasks. In Figure 4, the model receives a relational query, trims a 35.5 s clip to the precise 15.8–20.3 s interval that contains the described arrangement, draws a tight 2D box around the cabinet, propagates the cue with SAM2, lifts the mask into a point cloud, and produces a chain-of-thought that explicitly recounts the vertical stacking order. Figure 5 tackles a more abstract spatial question. Scene-R1 reasons that the door's location implies the camera-wearer sits on the left side of the long table; 12 o'clock relative to the reference chair points directly ahead, so the answer is the table. The point cloud shows the corresponding 3D scan. Together, these cases show how Scene-R1 yields transparent and accurate 3D scene understanding across both grounding and question answering tasks.

# C    THE USE OF LARGE LANGUAGE MODELS (LLMS)

During the final revision stages of this manuscript, we made selective use of Large Language Models (LLMs) to assist with language polishing in certain sections. Their application was specifically aimed at refining phrasing and ensuring grammatical accuracy to improve the overall clarity and readability of our work. All scientific contributions, methodologies, and core arguments were conceived and written by the authors, who retain full responsibility for the final content of this paper.

Language Query: A cabinet sits with a microwave under the top of it. it's got a small tv on top of it.

video length: 35.5s

**Stage 1**

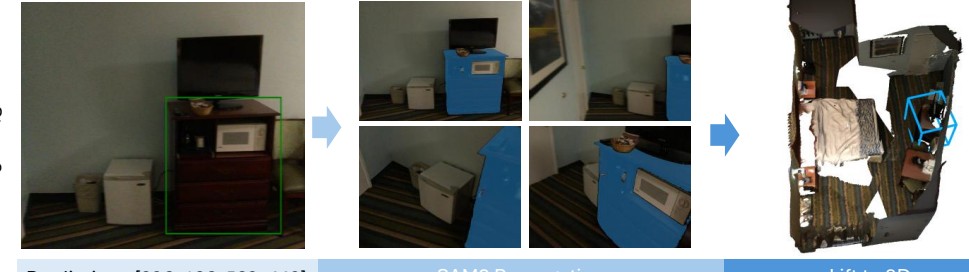

| Ground Truth: 14s – 22s | Prediction : 15.8s – 20.3s |

**Scene-R1 thinking:** The object description mentions a cabinet with a microwave under the top and a small TV on top. This matches the furniture arrangement seen in the video at around 20.0 seconds. The camera pans across the room, showing a wooden cabinet with a microwave on top and a flat-screen TV placed on the cabinet. This matches the description provided.

**Stage 2**

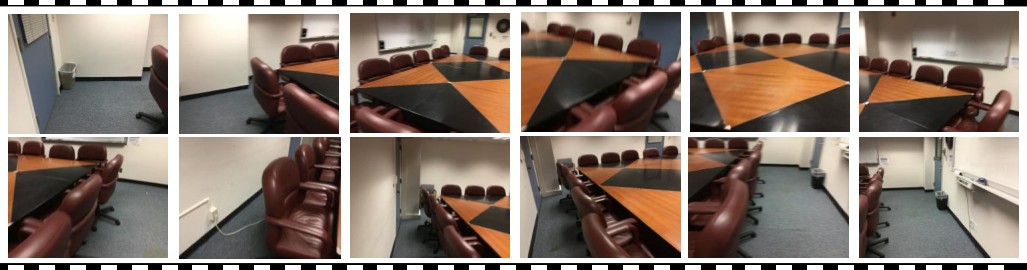

| Prediction : [326, 186, 583, 449] | SAM2 Propagation | Lift to 3D |

**Scene-R1 thinking:** The description suggests that there is a cabinet with a microwave under its top and a small TV on top of it. In the image, there is a cabinet-like piece of furniture with a microwave placed under its top surface and a television on top of it. The cabinet appears to be a piece of furniture with drawers, which fits the description of a cabinet.

Figure 4: Visualization of 3D visual grounding.

**Question:** There are a few chairs in a row and I am sitting on one of them. There is no more chair on my left and the door is to my right in the other side of the room. What is on the 12 o'clock of the chair that is on my right?

**Scene-R1 thinking:** To determine what is on the 12 o'clock of the chair that is on my right, I need to consider the orientation of the room and the relative positions of the objects. Since the door is to my right in the other side of the room, it implies I am sitting on the left side of the long table. The 12 o'clock position of an imaginary clock on the chair that is on my right means the 12 o'clock is directly in front of that chair. In this room, the long table is the dominant object, and the 12 o'clock position on the other side of any chair would logically be the left side of the table.

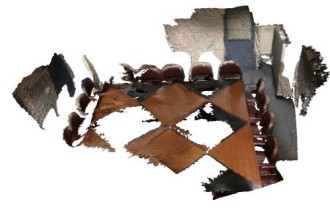

**Answer:** Table

Figure 5: Visualization of 3D visual question answering.

