# OpenReview forum: "Scene-R1: Video-Grounded Large Language Models  for 3D Scene Reasoning without 3D Annotations"
_ICLR.cc/2026/Conference — ICLR 2026 Conference Withdrawn Submission_

### Official Review · Reviewer_2xZ7 · 2025-10-27

**Soundness:** 2
**Presentation:** 2
**Contribution:** 2
**Rating:** 2
**Confidence:** 5

**Summary:**

This paper introduces **Scene-R1**, a video-grounded vision-language model (VLM) for 3D scene understanding that operates **without point-wise 3D annotations**. The core innovation lies in a **two-stage reinforcement learning** pipeline: temporal grounding followed by image grounding, both guided by lightweight rewards such as IoU and format compliance. Additionally, the paper extends the **GRPO** approach to 3D understanding by introducing an *exact-match reward*, achieving performance comparable to 3D LLMs that rely on point cloud inputs. Qualitative results further demonstrate the effectiveness of the model’s reasoning process.

**Strengths:**

1. The method removes the need for 3D point-wise instance labels while maintaining competitive performance under weak supervision.
2. By explicitly outputting chain-of-thought (CoT) reasoning, Scene-R1 improves interpretability compared with previous 3D LLMs, aligning with the growing emphasis on model transparency and explainability.
3. The two-stage RL structure (temporal then spatial grounding) provides flexibility and task generality across different 3D understanding tasks.

**Weaknesses:**

1. **The performance against other 3D LLMs remains limited**. The comparison with **VLM-Grounder** is not entirely fair, as it is a training-free agent and the reported results are based on only a 250-sample subset. For a more rigorous evaluation, performance should be assessed on the same benchmark samples used by VLM-Grounder. Although the paper claims that the proposed method does not require instance masks, the distinction between bounding-box-based and segmentation-based supervision is largely mitigated by the use of pretrained **SAM**. Moreover, the baseline **LLaVA-3D** similarly does not depend on pre-extracted 3D bounding boxes or per-instance segmentation, and should therefore be regarded as a **direct and comparable baseline** to the proposed approach.
2. **Similar grounding method:** The concept of back-projecting SAM masks to obtain 3D bounding boxes is not novel. The authors do not clearly distinguish their method from prior approaches such as **VLM-Grounder**.
3. **Limited benchmarks:** The RL framework is introduced not only for transparency but also for generalization. However, the evaluation is restricted to in-domain datasets. Cross-dataset evaluations on **Nr3D [1]**, **Multi3DRefer [2]**, or **Video-MME [3]** are encouraged to validate generalization.
4. **3DVQA implementation:** The paper claims that Scene-R1 can be fine-tuned for 3D-VQA tasks (L272). However, neither the training data nor the evaluation includes 3DVQA datasets such as **ScanQA [4]**, **SQA [5]**, or **MMScan [6]**. Since **VSI-Bench** does not provide a training set, it is unclear what data were used.
5. **Efficiency concerns:** The proposed multi-stage grounding combined with a DeepSeek-R1-style reasoning process substantially reduces efficiency. Ablation results show that the thinking process yields only marginal performance gains, casting doubt on the overall effectiveness of the proposed method.

[1] https://www.ecva.net/papers/eccv_2020/papers_ECCV/papers/123460409.pdf

[2] [[2309.05251\] Multi3DRefer: Grounding Text Description to Multiple 3D Objects](https://arxiv.org/abs/2309.05251)

[3] [[2405.21075\] Video-MME: The First-Ever Comprehensive Evaluation Benchmark of Multi-modal LLMs in Video Analysis](https://arxiv.org/abs/2405.21075)

[4] [[2112.10482\] ScanQA: 3D Question Answering for Spatial Scene Understanding](https://arxiv.org/abs/2112.10482)

[5] [[2210.07474\] SQA3D: Situated Question Answering in 3D Scenes](https://arxiv.org/abs/2210.07474)

[6] [[2406.09401\] MMScan: A Multi-Modal 3D Scene Dataset with Hierarchical Grounded Language Annotations](

**Questions:**

1. How does RL fine-tuning on grounding tasks improve performance on **VSI-Bench**? What prompts are used during VSI-Bench evaluation?
2. What is the **ablation setting**? The reported ablation results seem inconsistent with the main table. Additionally, what supervised fine-tuning (SFT) configuration is used in these ablations?
3. In L141, the authors state:
    *“We exploit this prior to teach it to understand the 3D world and minimize the amount of task-specific reinforcement learning (RL). The same architecture is used in all tasks optimized with GRPO.”*
    How exactly is the amount of task-specific RL minimized, given that the method introduces several task-specific rules, such as temporal and image grounding?
4. What do the **failure cases** look like? The paper presents only successful examples. A detailed failure mode analysis would provide deeper insight into the limitations of the proposed approach.

---

> ### Comment · Reviewer_2xZ7 · 2025-11-27
>
> In response to the general response from the authors: although the authors point out that 2D bounding boxes are easier to obtain, this justification is not well supported by the current training setup for 3D grounding datasets. The annotation process for these datasets still requires substantial 3D information (for example, the 3D web-based UI used in ScanRefer).
>
> It would be more appropriate to train on datasets designed for video object grounding/3D understanding, such as Cambrian-S (VSI-590K), which the authors mentioned in the rebuttal and then evaluate zero-shot performance on 3D understanding benchmarks. I understand, however, that conducting such experiments during the rebuttal phase is difficult.
>
> The authors also mention that LLaVA-3D uses 3D annotations for 3D object captioning. However, since 3D object captioning is not evaluated in the paper, this point does not provide clear support for the authors’ argument.
>
> I agree with many of qBeW’s comments and will consider raising my score if the authors can engage further in this discussion.

---

### Official Review · Reviewer_63FQ · 2025-10-31

**Soundness:** 3
**Presentation:** 2
**Contribution:** 2
**Rating:** 6
**Confidence:** 3

**Summary:**

The paper introduces a video-grounded large vision-language model (VLM) that performs 3D scene reasoning and grounding without any point-wise 3D annotations. Instead of relying on pretrained 3D detectors, Scene-R1 integrates reinforcement-learning-driven reasoning (R1-style) with a two-stage grounding pipeline, enabling transparent, interpretable, and annotation-efficient 3D reasoning.

The proposed method, Scene-R1, builds on Qwen2.5-VL-7B and is fine-tuned using GRPO. In Stage 1 (Temporal Grounding), the model reasons over video sequences to identify the most relevant temporal segment corresponding to a textual query. In Stage 2 (Image Grounding), it localizes the target object in selected frames by predicting 2D bounding boxes, accompanied by explicit chain-of-thought explanations. These 2D predictions are then lifted to 3D using depth maps and refined via a zero-shot segmentation module, producing accurate 3D localizations without any 3D supervision.

**Strengths:**

1. Annotation Efficiency:  Scene-R1 achieves competitive 3D reasoning and grounding performance without relying on dense point-wise 3D annotations or pretrained 3D detectors, greatly reducing the data and labeling cost.

2. The authors conducted comprehensive experiments with various existing works, and shows good performance.

**Weaknesses:**

1. The method rewards properly formatted CoT and task success (IoU/EM), but does not verify that the CoT is faithful to the internal decision path[1,2]

2. While the proposed pipeline has not been widely applied in existing 3D LLMs, its design does not represent a substantial conceptual departure from established video-grounding or multi-stage reasoning frameworks. The contribution feels more like an adaptation of existing ideas to a new input modality rather than a fundamentally novel approach.


[1] Sarkar, Advait. "Large language models cannot explain themselves." arXiv preprint arXiv:2405.04382 (2024).

[2] Kambhampati, Subbarao, et al. "Stop Anthropomorphizing Intermediate Tokens as Reasoning/Thinking Traces!." arXiv preprint arXiv:2504.09762 (2025).

**Questions:**

Please address the weakness mentioned above.

---

### Official Review · Reviewer_KQez · 2025-11-01

**Soundness:** 3
**Presentation:** 3
**Contribution:** 3
**Rating:** 6
**Confidence:** 4

**Summary:**

This paper introduces Scene-R1, a framework for 3D scene reasoning that operates directly on RGB-D video streams and, critically, requires no 3D point-wise annotations for training. The method uses a two-stage, VLM-based pipeline: (1) temporal grounding to select relevant video snippets and (2) image grounding to predict 2D bounding boxes. These 2D predictions are then lifted to 3D using SAM2 and a refinement module. The entire pipeline is optimized using reinforcement learning (GRPO), which both trains the model using lightweight 2D/textual rewards and encourages the generation of explicit chain-of-thought rationales for interpretability. The model is evaluated on 3D visual grounding, affordance grounding, and VQA, demonstrating competitive performance against other annotation-free baselines.

**Strengths:**

1. The most significant strength is the "annotation-free" nature of the 3D instance labeling. By learning from 2D bounding boxes and textual labels, the method drastically lowers the supervision requirements for 3D scene understanding, making it more scalable.
2. The integration of R1-style reinforcement learning to produce explicit chain-of-thought rationales adds a strong interpretability component, which is lacking in most 3D-aware LLMs.
3. The quantitative results are solid, showing that Scene-R1 outperforms other annotation-free baselines on several benchmarks (ScanRefer, SceneFun3D, VSI-Bench), validating the effectiveness of the proposed approach.

**Weaknesses:**

1. The system's design is a complex pipeline of multiple, powerful, pre-trained models (Qwen2.5-VL, SAM2, and a module inspired by SAI3D). This makes it difficult to ascertain how much of the strong performance is attributable to the novel RL framework versus the inherent power of these individual components.
2. The method's reliance on ground-truth depth ($D_t$) and camera poses ($T_t$) is a significant assumption. This data is not available in general "in-the-wild" videos and is the same data required to create the point clouds for detector-based methods. This weakens the claim of "bypassing 3D scene reconstruction" and limits the method's applicability to settings where a full 3D capture setup is already available.
3. The 2D-to-3D lifting process has several stages (2D box prediction, SAM2 segmentation, depth-based back-projection, refinement). This multi-step process seems susceptible to cascading errors, where a poor 2D box from the VLM could lead to an irrecoverably bad 3D localization.

**Questions:**

1. How critical is the explicit depth channel ($D_t$) and ground-truth pose ($T_t$)? What is the performance degradation if the model is run on RGB-only video and must rely on estimated depth/pose, or if it must operate without them? This seems to be the key bottleneck for real-world application.

---

### Official Review · Reviewer_qBeW · 2025-11-01

**Soundness:** 1
**Presentation:** 3
**Contribution:** 2
**Rating:** 2
**Confidence:** 4

**Summary:**

The paper proposes a video-grounded LLM which do not use 3D instance annotations for training. Specifically, the input to the model is a video: the VLM is asked to predict the relevant frames and then ground relevant object in this relevant portion of the video. For training these modules, GRPO losses are used. Next, these 2D predictions are lifted to 3D — for this, each predicted mask is tracked across frames using SAM-2 and then the resulting masks from all frames are fused in 3D. Next, another merging strategy from a prior work (SAI3D) is used to obtain a sharper mask. This becomes the prediction of the model. The paper compares its methods with prior methods that utilize 3D supervision as well as methods that do not use 3D supervision. The paper claims better performance than methods that do not use 3D supervision. The ablations show that RL training and thinking help improve performance over supervised fine-tuning.

**Strengths:**

- The paper is well-written and easy to follow
- The premise of training models without 3D supervision is interesting; additionally exploring RL training for these models is interesting as well.

**Weaknesses:**

- A big claim of the paper is that their method do not use 3D annotations. However, I think that is not entirely true — in “image grounding” task, the proposed model trains for supervising the mask prediction of the relevant object for each image in the video. This requires two kinds of supervision: a) “grounding” supervision which tells the model which object it should be grounding. b) “mask supervision” of that object across ALL video frames. These labels in scannet come from projecting the GT 3D segmentation masks to 2D. I will further argue that 3D mask annotations and 2D video masks are equivalent supervision for a posed RGB-D video i.e. either of these can be obtained from the other one via 2D projection or 3D unprojection. Hence, either of these supervisions is equally costly or inefficient. Hence, the claim that this method trains without 3D annotation appears wrong to me
- In the same vein, the comparisons in table-1 are potentially unfair:
    - In the “free from 3D instance or annotation supervision” section where the proposed method groups itself, the other baselines like vlm-grounder, open-scene and lerf do not use ANY supervision — neither any grounding supervision nor any mask supervision. The current method uses both these ground truth supervision as I argue in the first point
    - In the fully supervised methods section, the baselines are significantly old. Current SOTA is UniVLG (https://arxiv.org/abs/2503.10745) an the authors can check Table-1 of UniVLG for additional recent baselines.
-  “This architecture uniquely enables end-to-end reasoning directly on video streams, bypassing the need for offline 3D scene reconstruction”: This is a statement made in the introductions, however,  I think section 4.3 which lifts the 2D masks to 3D uses the reconstructed point clouds, and so do all the evaluations that follow in Table-1.

**Questions:**

The main question in my review, as I explain in the weakness section, is that the claim of not using 3D annotations seems false and the comparisions with zero-shot methods unfair. Any clarification would help here.

---

> ### Author Response · Authors · 2025-11-14
> **Response to Reviewer qBeW**
>
> We sincerely thank Reviewer qBeW for the detailed feedback. We appreciate the opportunity to provide clarification, especially regarding the critical distinctions in annotation cost, architectural contribution, and SOTA conventions, which we believe address the core concerns of the review.
>
> 1. Response to Weakness 1 (Regarding "false claim: no 3D annotations")
>
> We respectfully disagree with the assessment that our 2D reward signals are equivalent supervision or equally costly or inefficient to the point-wise 3D labels we avoid. This point is central to our contribution.
>
> Our primary claim is architectural: our framework is the first to explicitly remove the dependency on 3D detector priors. SOTA methods like LLaVA-3D (Zhu et al., 2024) and SeeGround (Li et al., 2025) explicitly require these 3D-detector modules to generate proposals. These 3D detector modules, in turn, must be trained on the very expensive, point-wise 3D instance segmentation masks that we successfully avoid.
>
> Scalability of 2D Rewards: Our method is built on the premise that coarse 2D box rewards are far more scalable and cheaper to acquire than dense 3D point-wise labels. As we clarify in our General Response, using 3D-to-2D projected data is a standard SOTA practice for sourcing these lightweight signals, also used by VLM-Grounder (Xu et al., 2024) .
>
> Proof of Learning: Our experiments demonstrate that our model can learn robust 3D knowledge from these coarse 2D signals. Our ablation study (Table 4) shows our GRPO framework yields a +38.2% absolute gain over the zero-shot backbone, proving the effectiveness of training on these lightweight rewards.
>
> 2. Response to Weakness 2 (Regarding "unfair comparison")
>
>  We agree that our Table 1 grouping was ambiguous. Our intent was to group methods by their dependency on 3D detectors, not their zero-shot status. We acknowledge this and will revise the table titles in the final version to explicitly label VLM-Grounder (Zero-Shot). We also note that VLM-Grounder relies on the closed-source GPT-4o API, whereas our method's SOTA results are achieved on an open-source 7B model, demonstrating our framework's efficiency.
>
> On UniVLG: Thank you for the suggestion. UniVLG is a relevant SOTA for point-cloud-input models. Our Scene-R1 operates on a different and challenging paradigm: operating directly from video streams. We will add a discussion of UniVLG to our related work section in the final paper.
>
> 3. Response to Weakness 3 (Regarding "bypassing 3D reconstruction" and Sec 4.3)
>
> You raise a profound point that highlights the core novelty of our architecture. We apologize for the imprecise phrasing and clarify this in two parts:
>
> On "Bypassing Reconstruction": We apologize for the imprecise phrasing "bypassing... reconstruction". Our intended meaning was "bypassing the need for a pre-built 3D point cloud as model input," which is a significant bottleneck for methods like LLaVA-3D and SeeGround. Our use of RGB-D streams and camera poses (to lift 2D predictions and for evaluation) is indeed the standard experimental setup for this domain, consistent with VLM-Grounder (Xu et al., 2024) .
>
> On the Purpose of Sec 4.3 (Our SAI3D Refinement): This is the key distinction. SOTA methods (e.g., LLaVA-3D, Video-3D LLM) use 3D detectors like Mask3D as a front-end to generate 3D proposals. The LLM's task is merely to select from these pre-generated 3D candidates.
>
> Our Scene-R1 does the opposite. Our framework has no 3D detector front-end. Our RL-VLM autonomously generates its own coarse 2D prediction from the video stream.
>
> We use the SAI3D-inspired module only as a back-end refinement step. Its purpose is to take our own noisy 2D prediction and clean it up by enforcing multi-view geometric consistency, yielding a precise 3D box. This is fundamentally different from relying on a 3D detector for proposals.
>
> We hope these clarifications, especially the distinction between lightweight 2D rewards and expensive point-wise 3D detector training, address your primary concerns. If you have any further questions, we welcome discussion.
>
> [1] LLaVA-3D (Zhu et al., 2024) Zhu, C., Wang, T., Zhang, W., Pang, J., & Liu, X. (2024). Llava-3d: A simple yet effective pathway to empowering lmms with 3d-awareness. arXiv preprint arXiv:2409.18125.
>
> [2] SeeGround (Li et al., 2025) Li, R., Li, S., Kong, L., Yang, X., & Liang, J. (2025). Seeground: See and ground for zero-shot open-vocabulary 3d visual grounding. In Proceedings of the Computer Vision and Pattern Recognition Conference (pp. 3707-3717).
>
> [3] VLM-Grounder (Xu et al., 2024) Xu, R., Huang, Z., Wang, T., Chen, Y., Pang, J., & Lin, D. (2024). Vlm-grounder: A vlm agent for zero-shot 3d visual grounding. arXiv preprint arXiv:2410.13860.
>
> [4] Video-3D LLM (Zheng et al., 2025)Zheng, D., Huang, S., & Wang, L. (2025). Video-3d llm: Learning position-aware video representation for 3d scene understanding. In Proceedings of the Computer Vision and Pattern Recognition Conference (pp. 8995-9006).

---

> > ### Comment · Reviewer_qBeW · 2025-11-17
> >
> > Thank you to the authors for their response. I responded to some of the points raised here in the common response. I add some additional comments here.
> >
> > > Our intended meaning was "bypassing the need for a pre-built 3D point cloud as model input," which is a significant bottleneck for methods like LLaVA-3D and SeeGround.
> >
> > I am still not sure if the new formulation is precise. This is what LLaVA-3D says in section 5.3: "Without relying on the constructed point cloud, our method could directly decode the accurate 3D bounding boxes from 3D patches and achieve the SOTA performance (49.8 Acc@0.25 on Multi3DRefer) in the single-stage manner."
> >
> > Overall, I see a big confusion in author's response where the assumption is that LLAVA-3D uses object proposals as input while they seem to not use it. They report two numbers in Table-6: one where they use detectors and the other where they do not use detectors.

---

### Author Response · Authors · 2025-11-13
**General Response: Clarifications on Core Contribution, SOTA Comparisons, and Framework Value**

We thank all reviewers for their time and insightful feedback. We note the clear divergence in scores (6, 6, 2, 2) and believe the lower ratings stem from several fundamental misunderstandings regarding our core contribution and its relationship to the current SOTA. We would like to clarify these points.

1. Clarification on LLaVA-3D and SOTA Dependencies (Re: qBeW W1, 2xz7 W1)

We must correct a critical misunderstanding regarding the SOTA baselines. Reviewers qBeW appear to believe that methods like LLaVA-3D (Zhu et al., 2024) do not rely on 3D detectors, which is factually incorrect.

The LLaVA-3D paper (Zhu et al., 2024) explicitly states in Sections 5.2 and 5.3 that its 3D grounding and captioning evaluations rely on an "off-the-shelf 3D instance segmentation model" to generate 3D object proposals.

This dependency is not an isolated case. As another example, SeeGround (Li et al., 2025) states in Section 3.1 that its method also explicitly requires an "open-vocabulary 3D detection framework" to pre-populate its Object Lookup Table (OLT).

2. Clarification on Our Core Architectural Contribution (Re: 2xz7 W1, qBeW W1)

This clarification directly highlights our primary contribution. The SOTA methods mentioned above still depend on 3D detectors/segmentors, which themselves require training on expensive, point-wise 3D dense labels.

Our Scene-R1 framework is the first to explicitly remove this dependency on 3D-detector priors. We instead leverage components like SAM2 (trained on massive 2D data) and SAI3D (a zero-shot module). This is a novel architectural shift towards a more scalable and annotation-efficient paradigm for 3D scene understanding.

3. Clarification on Table 1 Comparisons (Re: qBeW W2, 2xz7 W1)

Regarding the fairness of Table 1, our original intent was to distinguish methods based on their dependency on 3D-trained encoders and 3D data, not strictly "zero-shot" status.

We acknowledge that our grouping title (Free from 3D Instance...) was ambiguous and caused confusion. We will revise this in the final version to explicitly label zero-shot methods (e.g., VLM-Grounder (Zero-Shot)).

However, we must also note a key difference in experimental setup: VLM-Grounder (Xu et al., 2024) relies on the powerful, closed-source GPT-4o API. In contrast, our Scene-R1 achieves its results by fine-tuning a fully open-source, 7B-parameter model (Qwen2.5-VL-7B). We argue that demonstrating this capability on a small, open model is a significant contribution.

4. Clarification on Data Source and the Value of Our RL Framework (Re: qBeW W1, KQez W1, 2xz7 W5)

Finally, we address the concern about our 2D data source and the value of our framework.

We acknowledge that our 2D reward signals for ScanNet are derived from 3D-to-2D projection.

However, we must emphasize that the cost of obtaining these "coarse" 2D boxes is far lower than the point-wise 3D masks we avoid. This data pipeline is standard SOTA practice, also used by VLM-Grounder (Xu et al., 2024) and the recent Cambrian-S (Yang et al., 2025), which also uses 3D-derived annotations.

Concluding Remark on Input Assumptions (Re: qBeW W3, KQez W2)

We also apologize for the imprecise phrasing "bypassing the need for offline 3D scene reconstruction". Our intended meaning was "bypassing the need for a pre-built 3D point cloud as model input," which is a bottleneck for methods like LLaVA-3D and SeeGround. Our use of RGB-D streams and camera poses is a standard experimental setup, consistent with VLM-Grounder (Xu et al., 2024) and Cambrian-S (Yang et al., 2025).

References

Li, R., et al. (2025). "Seeground: See and ground for zero-shot open-vocabulary 3d visual grounding." Proceedings of the Computer Vision and Pattern Recognition Conference.

Xu, R., et al. (2024). "Vlm-grounder: A vlm agent for zero-shot 3d visual grounding." arXiv preprint arXiv:2410.13860.

Yang, S., et al. (2025). "Cambrian-S: Towards Spatial Supersensing in Video." arXiv preprint arXiv:2511.04670.

Zhu, C., et al. (2024). "Llava-3d: A simple yet effective pathway to empowering lmms with 3d-awareness." arXiv preprint arXiv:2409.18125.

---

> ### Comment · Reviewer_qBeW · 2025-11-17
>
> Thank you to the authors for their rebuttal. My responses are inline:
>
> > "We must correct a critical misunderstanding regarding the SOTA baselines. Reviewers qBeW appear to believe that methods like LLaVA-3D (Zhu et al., 2024) do not rely on 3D detectors, which is factually incorrect."
>
> I think the authors wrongly attributed this comment to me instead of reviewer 2x27. In anycase, my understanding is that LLAVA-3D uses 3D labels for supervision. They directly decode bounding boxes for the grounding tasks through their decoder, and then supervise it with GT 3D grounding labels. They DO NOT use object proposals as input to their model. See table-6 where they have seperate rows for single-stage 3D LLMs and two-stage 3D LLMs. Only their captioning results use off-the-shelf 3D object proposals as input to the model.
>
> Hence, I agree with reviewer 2x27 that llava-3d is a directly comparable baseline. The authors may argue that their method do not use '3D annotations' and thus llava-3d is not directly comparable, but as I articulate in my original review (and discussion below), I do not believe that the current version of the paper do not use '3D annotations'
>
> > However, we must emphasize that the cost of obtaining these "coarse" 2D boxes is far lower than the point-wise 3D masks we avoid. This data pipeline is standard SOTA practice, also used by VLM-Grounder (Xu et al., 2024) and the recent Cambrian-S (Yang et al., 2025), which also uses 3D-derived annotations.
>
> I do not understand this argument. If one uses 3D-derived annotations, I am not sure how it is accurate to claim "training without 3D annotations".
>
> > However, we must emphasize that the cost of obtaining these "coarse" 2D boxes is far lower than the point-wise 3D masks we avoid.
>
> Could you please explain further why this is the case? Infact, I will argue the opposite -- LLAVA-3D uses 3D bounding boxes for supervision (not 3D segmentation masks). Labelling one single 3D bounding box in 3D sounds easier than labelling consistently tracked 2D boxes across all images of the entire video through which the scene is created.

---

### Author Response · Authors · 2025-12-04

We sincerely thank all the reviewers and ACs for their time, effort and contribution.

---

### Note · Authors · 2025-12-04

I have read and agree with the venue's withdrawal policy on behalf of myself and my co-authors.